# Developing a Scale Using Item Response Theory of the Self-Resilience in Taekwondo Players

**DOI:** 10.3390/ijerph20010728

**Published:** 2022-12-30

**Authors:** Hyun-Bin Kim, Jung-Hun Nam, Dong-Hwa Chung, Eun-Hyung Cho

**Affiliations:** 1Department of Physical Education, Dae Duk University, Daejeon 34111, Republic of Korea; 2Department of Physical Education, Korea National Sport University, Seoul 05541, Republic of Korea; 3Department of Convergence of Sports, Sang Myung University, Cheonan-Si 31162, Republic of Korea; 4Department of Sport Science, Korea Institute of Sport Science, Seoul 01794, Republic of Korea

**Keywords:** validation, ego-resilience, taekwondo

## Abstract

The purpose of this study was to identify the ego-resiliency of Taekwondo athletes and to develop a scale measuring such skills. We collected preliminary data using an open-ended online survey targeting Taekwondo athletes from nine countries (South Korea, China, Malaysia, United States, Spain, France, Brazil, United Kingdom, and Taiwan) who participated in international competitions between 2019 and 2020. We extracted participants’ ego-resiliency from 48 survey responses, guided by expert meetings and a thorough literature review. We verified our Taekwondo ego-resiliency scale’s construct validity using 741 survey responses. We utilized V coefficients, parallel analysis, an exploratory structural equation model, maximum likelihood, confirmatory factor analysis, and multi-group confirmatory factor analysis for data analysis. We identified four core ego-resiliency types: “empathy,” “coach support,” “care,” and “parent support/effort”. Our final measure, which demonstrated evidence of reliability and validity, comprises 18 items spanning 4 factors, with each item rated on a 3-point Likert scale.

## 1. Introduction

Athletes are subject to relentless competition and training and often suffer from psychological challenges such as frequent frustrations and stress. Hence, the aspect of resilience among athletes is an essential attribute to demonstrate the best athletic performance. Previous studies in sports psychology have focused on effectively suppressing and regulating negative thinking or emotions as a means to overcome athletes’ perceived adversity and suffering. However, with a shift in the paradigm of psychology research from a pathology-focused view to a positivity-focused view [1,2], recent studies have focused on athletes’ positive attributes. These studies emphasize positivity in individuals and have pinpointed resilience as a key attribute of athletes facing adversity or risky situations [3,4,5].

The concept of resilience was initially proposed in natural sciences to describe the rate of change of special metals against a stressor [6,7], and it has been used as the universal standard for understanding the properties of metals. However, the concept has been adopted in other disciplines, including economics and psychology, to describe a social and psychological attribute, and is currently widely utilized in various disciplines. In psychology, resilience is referred to as ego-resiliency and is defined as a self-control-related factor to effectively overcome anxiety, frustration, or hardship [6,8]. Early studies on ego-resiliency classified the concept as a personality trait and explained it through individuals’ positive attitudes, goal orientation, emotional regulation, and social ability. However, in recent years, ego-resiliency has been conceptualized as the overall process of overcoming risk factors in the surrounding environment and promoting well-being [2,9] based on a dynamic view, where the attribute is shaped through an interaction with the surrounding environment, as opposed to being a character trait [10,11].

The new concept of ego-resiliency divides the concept into two dimensions—intrinsic and extrinsic—and defines the intrinsic dimension to encompass intellectual ability, affect, will, and spirituality, and the extrinsic dimension to deal with protection or assistance from the surrounding environment. Multiple instruments have been developed to measure ego-resilience, including the Adolescent Resilience Scale (ARS) [12], Baruth Protective Factors Inventory (BPFI) [13], Brief-Resilience Coping Scale (BRCS) [14], Connor-Davidson Resilience Scale (CD-RISC) [15], Resilience Scale for Adults (RSA) [16], Resilience Scale (RS) [17]. However, with the exception of the scale developed by Connor and Davidson (2003), all ego-resiliency scales were developed based on the early view that defines ego-resiliency as a set character trait and thus has limitations in measuring ego-resiliency viewed as an overall process, failing to provide an accurate empirical measurement of ego-resilience in individuals [10].

Studies on ego-resiliency in sports have primarily investigated the effects of ego-resiliency on sports activities [18] or the roles of ego-resiliency in sports behaviors [19], with studies examining athletes’ ego-resiliency relatively lacking. Past studies on athletes’ ego-resiliency [20,21] examined the effects of ego-resiliency in athletes using ego-resiliency scales developed on the general population. Further, the only available ego-resiliency scales for athletes are the ones developed by Kim and Kim (2016), Lee (2013), and Choi and Chang (2015) [22,23,24].

These studies are significant in terms of structuralizing the concept of ego-resiliency among athletes, but the scales also define ego-resiliency as an athlete’s personality trait based on the early conceptualization of ego-resiliency and fail to embody the concept of ego-resiliency as an overall process. Moreover, according to Kim (2016) [25] who analyzed the validity of these scales, these studies employed principal component analysis (PCA) and traditional tests indiscreetly during the identification of factor structures, which distorted—underestimated or overestimated—the factor structures [26,27] and led to inappropriate items being included in the scales for athletes. Thus, to define athletes’ ego-resiliency empirically and to develop a reliable ego-resiliency scale, athletes’ ego-resiliency must be understood from the perspective of an overall process, and different statistical techniques must be used [27,28,29,30].

Taekwondo is a global sport with 209 member nations (WTF, 2017) and adopted in 16 single international competitions and multi-sport events in 15 countries. It is a combat sport in which victory or defeat is determined in a match involving kicking and punching between two athletes. Because athletes experience a high level of stress and negative emotions during training or match, coping skills to deal with negative emotions and surrounding situations are as important as strong physical fitness and skills [31]. Due to such features of Taekwondo, there has been active research on the psychological coping skills of Taekwondo athletes [32,33]. However, studies pertinent to Taekwondo athletes’ negative emotions or psychological coping skills have primarily focused on examining sport psychological skills (SPS), with little research interest on ego-resiliency. Ego-resiliency is a more fundamental concept than SPS. It encompasses motive regulation and cognitive abilities to make adjustments while maintaining composure and not causing behavioral or emotional problems in a threatening environment, and it also includes the ability to adjust one’s control in accordance with the environmental demands [34]. Thus, an empirical examination of ego-resiliency is crucial to predict or understand Taekwondo athletes’ psychological equilibrium and ability to overcome hardship. However, there is no established scale that clearly conceptualizes ego-resiliency and accurately measures the construct in Taekwondo athletes.

Our review of past studies on ego-resiliency in sports and relevant scale development highlighted the need to define ego-resiliency in Taekwondo athletes as a process and to develop an appropriate scale to measure this concept in order to understand and predict the ways Taekwondo athletes overcome adversity and challenges in a sports setting. Thus, this study aims to conceptualize ego-resiliency in Taekwondo athletes and develop a scale using the unified framework of validity concept [35] and Rasch model for construct validity [36].

## 2. Methods

### 2.1. Research Subjects and Data Collection

The study population comprised experts and athletes. The expert group was involved in the translation of the scale and content validity evaluation, while the athlete group was involved in the construct validity evaluation phase. The expert group was recruited via purposive sampling in order to enroll professional sports psychologists.

As shown in Table 1, the expert group consisted of two sports psychology professors, two researchers with a Ph.D., two experts with more than 10 years of experience as Taekwondo national team coaches, and two Taekwondo athletes with more than ten years of career in the national team. Further, Table 2 the questionnaire to evaluate the construct validity of the Ego-Resiliency Scale for Taekwondo athletes was administered online to Taekwondo athletes from nine countries who competed in 2019–2020 international Taekwondo competitions (i.e., S. Korea, China, Malaysia, United States, Spain, France, Brazil, United Kingdom, Taiwan), with the cooperation of World Taekwondo. The questionnaire, translated into English, was designed as a self-report questionnaire, and the responses were collected immediately upon submission. Of the 754 questionnaires administered, 13 were excluded for missing responses or duplicate responses, resulting in a total of 741 questionnaires being included in the study. The collected questionnaires were randomly divided into two types (group A/group B) for analysis.

### 2.2. Analysis

As shown in Table 3, the analysis for this study was performed across three stages (substantive domain, structural domain, external domain) in accordance with the protocol for construct validation program proposed by Benson (1998) with reference to the Rasch model for unified validity.

In the substantive domain, literature was reviewed, and the scale items were developed and validated. In the structural domain, the factor structure for the Ego-Resiliency Scale was identified, and the reliability of the scale was evaluated. In the external domain, the power of the Ego-Resiliency Scale was tested.

### 2.3. Data Proce$ssing

In the substantive domain, the items of the Ego-Resiliency Scale were developed through a process of content validity evaluation using the V coefficient proposed by Aiken (1985) [37]. Testing of unidimensionality, evaluation of the appropriateness, and fit of the response categories using WINSTEPS 3.65 [38]. In the structural domain, exploratory structural equation modeling (ESEM) was performed using Mplus 7.4 [39]. In addition, confirmatory factor analysis (CFA) and reliability analysis were performed using ESEM. In the external domain, the power of the Ego-Resiliency Scale was analyzed using latent mean analysis with sex differences as the reference.

## 3. Results

### 3.1. Substantive Domain

#### 3.1.1. Formulation of Items for the Ego-Resiliency Scale

As shown in Table 4, the items for the Ego-Resiliency Scale for Taekwondo athletes were developed with reference to the studies by Connor and Davidson (2003) and Zautra, Hall, Stuart, and Murray (2010) that proposed the concept of ego-resiliency as an overall process.

Further, 48 items for 16 factors were developed based on the factors suggested in these studies: self-efficacy, problem-solving skills, emotion/impulse, empathy/acceptance, goal, optimism, search for meaning, spirituality, care relationship in the team, team dynamics, care relationship in the family, family dynamics, care relationship in community, dynamics in community activities, positive infections with colleagues, and prosocial expectations with colleagues. The items were developed in accordance with the protocol delineated by Crocker and Algina (1986) [40], and three items were developed for each factor with due consideration to the delivery of content, grammar, delivery of meaning, and positive content. Then, the content validity of the items was tested by an expert panel. Aiken’s (1985) V coefficient and binomial probability distribution for each item were calculated to ensure the objectivity of the content validity evaluation. The results indicated that items 47 and 48 were inappropriate for assessing ego-resiliency in Taekwondo athletes owing to the statistical insignificance (>0.5) of the V coefficient. Hence, these two items were removed.

#### 3.1.2. Assessment of Unidimensionality

To apply the Rasch model, unidimensionality—a prerequisite—was tested. Unidimensionality was analyzed using principal component analysis (PCA) on WINSTEPS 3.65 [38], and as shown in Table 5, the explained variance was 48.3%, satisfying the criteria for unidimensionality (observed variance ≥ 20%) (DeMars, 2010) [41]. Thus, the items in the Ego-Resiliency Scale were confirmed to be unidimensional.

#### 3.1.3. Suitability of Rating Scale (Response Categories)

As shown in Table 6, we used the rating scale analysis in the Rasch model to determine the optimal rating scale. The suitability of the rating scale was assessed with reference to the category probability curve of each category with the following criteria: a minimum of ten counts per category, frequency (%) distribution for each category, average measure (AM), standardized infit and outfit (7.5–1.30) [42] and change of step calibration (SC). On the basis of these criteria, the suitability of 5-point, 4-point, and 3-point Likert scales was analyzed, and the 5-point and 4-point Likert scales did not meet the criteria. The absolute value of the SC was smaller than 1.4 for the 3-, 4-, and 5-point scales, falling short of the criteria of 1.4–5.0. However, the 3-point Likert scale (11,223) had at least ten counts and even frequency across categories (%). Further, the AM increased with an increasing level of category. The infit and outfit values of each category were within the range of 7.5–1.30 [42], and the absolute value of the SC was also within the range of 1.4–5.0. Thus, a 3-point Likert scale was found to be suitable for the Ego-Resiliency Scale for Taekwondo athletes.

#### 3.1.4. Item Fit

The RSM (Andrich, 1978) was used to assess the fit of items of the Ego-Resiliency Scale. Mean square fit statistic (MNSQ) and point-biserial correlation (PBC) were used as the fit indices.

Item fit indicates the discriminatory power of each item of the Ego-Resiliency Scale and was assessed using MNSQ as the criterion. The MNSQ criterion was set to a range of 0.75–1.3. As shown in Table 7, 14 items did not meet the MNSQ criterion, indicating that these items are inappropriate for assessing ego-resiliency in Taekwondo athletes.

### 3.2. Structural Domain

#### 3.2.1. Exploratory Structural Equation Modeling (ESEM)

The number of factors for the Ego-Resiliency Scale for Taekwondo athletes was determined based on root mean square error of approximation (RMSEA) and interpretability [41,43,44].

As shown in Table 8, the RMSEA dropped to below 0.8 from four factors, and the difference in RMSEA value was smaller than 0.01 from four factors to six factors. Thus, the number of factors was set to four [45].

As shown in Table 9, ESEM was performed based on a four-factor structure. In the ESEM, items with multidimensionality (statistically significant factor loading of 0.2 or higher onto two or more factors) [46] and items that have been allocated to a factor completely different from that in the theoretical model and thus hinder a realistic interpretation [47] were deleted. A total of 28 items were deleted through this process.

#### 3.2.2. CFA Using Maximum Likelihood

CFA was performed using maximum likelihood (ML) to verify the factor structure of the Ego-Resiliency Scale for Taekwondo athletes. As shown in Table 10, the model had an acceptable fit with an RMSEA (criterion: <0.08) of 0.067 and a Tucker-Lewis Index (TLI; criterion: >0.90) of 0.912.

Based on ESEM and CFA, the Ego-Resiliency Scale for Taekwondo athletes was finalized to four factors with 18 items. Each factor was named based on the contents of its items. The first factor was named “Empathy” since it contained items about “I try to understand a colleague’s thoughts and feelings,” “I feel pain when a colleague or friend is hurt.” The second factor was named “Support from coach since it contained items about “I have a coach who encourages me,” “I have a coach who wants me to do my best,” and “I have a coach who can help me.” The third factor was named “Care” since it contained items about “I undertake important roles in the team,” “I get involved in activities to help colleagues or people who are having a hard time,” and “I frequently present my opinions for the good of the team.” The fourth factor was named “Parental support and personal endeavor” since it contained items about “I consult my parents,” “My parents are highly interested in my sports career and school life,” “My parents detect my emotions easily,” “I try hard to realize my dream,” “I always try hard to achieve my goals or plans.”

Next, the concurrent and discriminant validities of the Ego-Resiliency Scale were analyzed. The concurrent validity was established, as the criteria for average variance extraction (criterion: >0.50) and construct reliability (CR; criterion: >0.70) were met. The discriminant validity was also established, as the square of the correlation coefficient between the two factors with the strongest correlation (0.4982 = 0.248; support from leadership and parental support and personal endeavors) was smaller than the average variance extracted (AVE) of these factors (Woo, 2012).

#### 3.2.3. Reliability and Difficulty of Each Factor

The reliability of the Ego-Resiliency Scale for Taekwondo athletes was tested with reference to the response and item reliabilities.

As shown in Table 11, the index of separation was larger than 2.0, and reliability was above 0.80, with infit and outfit within the range of 0.75–1.3 [41], thereby indicating good item and response reliabilities. These results suggest that the Ego-Resiliency Scale is a reliable scale for assessing ego-resiliency in Taekwondo athletes and that it can accurately assess the level of ego-resiliency in these athletes.

### 3.3. External Domain

#### Multiple Group CFA (MCFA)

In the external domain, MCFA was performed to verify the power of the Ego-Resiliency Scale. Measurement invariance was tested using MCFA based on sex differences. As shown in Table 12, the significance of ∆X^2^ of configural equivalence and metric equivalence was 0.111, which is larger than 0.05, at a 95% confidence level, showing that these are not significant. These results indicate that the power of the Ego-Resiliency Scale for Taekwondo athletes does not vary according to the athletes’ sex, confirming the validity of the scale for assessing ego-resiliency in Taekwondo athletes.

## 4. Discussion

This study aimed to develop a scale for assessing ego-resiliency—an emerging concept in the field of sports psychology. The concept structure of ego-resiliency in an overall perspective was [15] first identified based on a review of literature [9], and the construct was categorized via domain-referenced testing through expert discussions. Ego-resiliency in Taekwondo athletes was categorized into self-efficacy, problem-solving skills, emotion/impulse, empathy/acceptance, goal, optimism, search for meaning, spirituality, care relationship in team, team dynamics, care relationship in family, family dynamics, care relationship in community, dynamics in community activities, positive infections with colleagues, and prosocial expectations with colleagues. In addition, the items for the scale were generated in accordance with the method proposed by Crocker and Algina (1986) in consideration of negative items, the accuracy of the item, grammar, and alignment with the goal of the test. The content validity of the items was evaluated objectively without being influenced by researchers’ and experts’ personal views and biases by using Aiken’s (1985) V coefficient.

In general, eliminating researchers’ views during content validation of generated items in the development of a psychological scale is highly challenging. Moreover, this fact substantially hinders establishing sound theoretical grounds for the scale [37]. Thus, objective indices such as the V coefficient should be used to verify the content validity of developed items without error.

In the present study, the unified framework of validity concept [35] and construct validity program using the Rasch model [36] were used to develop the Ego-Resiliency Scale for Taekwondo athletes. Accordingly, ESEM was performed to identify the suitable factor structure. One key benefit of ESEM is that it enables the identification of the optimal factor structure in consideration of interpretability by providing the significance and effect sizes of each item [26,43,46].

The factor loadings of each item were assessed with reference to the statistical significance criterion proposed by Jennrich (2007) [48], and items with a significant cross-loading with a factor loading of 0.20 or higher for two or more factors were deleted repeatedly. As a result of this process, a four-factor structure with 18 items was determined for the Ego-Resiliency Scale, and this factor structure was verified through CFA using ML.

CFA confirmed that the Ego-Resiliency Scale has good concurrent validity and discriminant validity without having to delete more items, so the Ego-Resiliency Scale for Taekwondo athletes was finalized with four factors and 18 items.

The reliability of the scale was also tested, and the item and response reliabilities were all above 0.80, showing that the Ego-Resiliency Scale is both valid and reliable in assessing ego-resiliency in Taekwondo athletes.

Finally, the power of the Ego-Resiliency Scale was assessed by latent means analysis with reference to sex, a demographic characteristic. The results confirmed that the scale has an invariant power across the two sexes among Taekwondo athletes.

Through this process, the Ego-Resiliency Scale for Taekwondo athletes was finalized into four factors: empathy, support from leadership, care, parental support, and personal endeavors. These results are similar to the findings of Lee and Cho (2005) and Zautra, Hall, Stuart, and Murray (2010) who conceptualized ego-resiliency. These authors have stated that an individual’s ego-resiliency is not a personality trait but an outcome of bonding and empathizing, as well as mutually supportive relationships among family and members of social organizations. Our results also showed that Taekwondo athletes’ ego-resiliency is shaped by support from parents and leadership, as well as empathy and care shared with colleagues and friends. In essence, Taekwondo athletes’ ego-resiliency is formed through empathy and acceptance from parents and leadership manifested as active support, and through empathy and considerate attitudes among colleagues.

In addition, on the basis of the fact that ego-resiliency is a self-control factor that enables individuals to effectively overcome adversity or hardship [8], fostering supportive, empathetic, and considerate relationships with Taekwondo athletes would be an important practical measure to enhance their psychological coping abilities.

## 5. Conclusions and Recommendations

This study aimed to develop the Ego-Resiliency Scale for Taekwondo athletes with reference to the concept of ego-resiliency as an overall process, and the following conclusions were drawn. First, Taekwondo athletes’ ego-resiliency was categorized into empathy, support from leadership, care, parental support, and personal endeavors. Second, the Ego-Resiliency Scale for Taekwondo athletes comprises four factors with 18 items, each rated on a three-point Likert scale. The scale had acceptable reliability and power. We present the following recommendations for future studies: There is an inadequate body of research data on the contribution of ego-resiliency on the qualitative aspect of athletic careers or Taekwondo athletes. Thus, subsequent studies should employ the Ego-Resiliency Scale to examine the roles and functions of ego-resiliency in the face of frustrations and adversities encountered by Taekwondo athletes in highly competitive situations.

## Figures and Tables

**Table 1 ijerph-20-00728-t001:** General characteristics of the expert group.

Group		Gender	Number
Experts	Professor/doctor in sport psychology	Male	2
		Female	1
	Coach	Male	2
		Female	1
	Athlete	Male	1
		Female	2

**Table 2 ijerph-20-00728-t002:** General characteristics of survey participants.

Domain		N	%	Domain		N	%
Group A (371) Average age 22.3	Male	235	63.6	Group B (370) Average age 23.7	Male	228	61.6
	Female	136	36.4		Female	142	38.4
	Less than 4 yrs	6	1.7		Less than 4 yrs	9	2.1
	5 to 7 yrs	86	23.3		5 to 7 yrs	95	25.7
	8 to 10 yrs	224	60.4		8 to 10 yrs	221	59.7
	More than 11 yrs	55	14.6		More than 11 yrs	46	12.5
	S. Korea	52	22.1		S. Korea	47	20.8
	UK	37	15.7		UK	39	17.1
	US	39	16.6		US	39	17.1
	Malaysia	21	8.9		Malaysia	28	12.8
	China	39	16.6		China	29	12.9
	Spain	22	9.4		Spain	19	8.3
	France	17	7.2		France	23	10.1
	Taiwan	8	3.5		Taiwan	4	0.9

**Table 3 ijerph-20-00728-t003:** Construct validity analysis.

Domain	Analysis Method	Data Source
Substantive	Literature review	Open-ended Survey
	Inductive categorization/item development according to theoretical model/content validity verification	
	Unidimensionality verification Response category verification Conformity verification	Group A
Structural	Exploratory structural equation	
	Confirmatory factor analysis	Group B
	Reliability analysis	
External	Latent mean analysis	

**Table 4 ijerph-20-00728-t004:** Items assessing Taekwondo athletes’ sport psychological skills.

Item	V Coefficient	Item	V Coefficient	Item	V Coefficient
1	0.77 **	21	0.76 **	41	0.29 **
2	10.33 **	22	0.87 **	42	0.43 **
3	0.66 *	23	0.49 **	43	0.31 **
4	0.55 *	24	0.84 **	44	0.57 **
5	0.54 **	25	0.92 **	45	0.43 *
6	0.47 **	26	0.52 **	46	0.82 **
7	0.57	27	0.62 **	47	0.11
8	0.49 **	28	0.47 **	48	0.07
9	0.73 **	29	0.92 **		
10	0.80 **	30	0.83 **		
11	0.94 **	31	0.64 **		
12	0.90 **	32	0.57 **		
13	0.87 **	33	0.66 **		
14	0.93 **	34	0.49 **		
15	0.55 **	35	0.43 **		
16	0.87 **	36	0.27 **		
17	0.79 *	37	0.44 **		
18	0.90 **	38	0.85 **		
19	0.88 **	39	0.78 **		
20	0.88 **	40	0.90 **		

* 0.05, ** 0.01.

**Table 5 ijerph-20-00728-t005:** Verification of unidimensionality.

		Eigenvalue	%

Explained variance	Person	26.8	29.8
	Item	16.6	18.5
Unexplained variance		46.3	51.7
Total variance		89.7	100

**Table 6 ijerph-20-00728-t006:** Verification of the 3-point Likert scale’s appropriateness.

	Count	%	AM	Infit	Oufit	SC	[SC]
1	447	4	−0.70	0.94	0.92	-	
2	1202	13	10.02	0.86	0.86	−10.43	10.43
3	3002	28	0.80	10.10	10.08	−0.89	0.54

**Table 7 ijerph-20-00728-t007:** Results of item relevance verification.

Item	LOGIT	MNSQ		PBC	Item	LOGIT	MNSQ		PBC
		Infit	Outfit				Infit	Outfit	
1	1.55	2.04	2.22	0.45	18	1.22	0.97	0.93	0.47
7	1.47	1.99	2.00	0.52	14	−0.56	0.21	0.65	0.59
3	−0.82	1.81	1.79	0.34	20	−0.40	0.20	0.23	0.51
29	0.98	1.67	1.74	0.47	23	−0.36	0.78	0.77	0.42
34	−0.95	1.51	1.59	0.44	46	−0.17	0.71	0.70	0.40
44	−0.44	1.40	1.44	0.57	25	−0.05	0.69	0.68	0.39
31	1.22	1.50	1.47	0.47	32	−0.52	0.68	0.65	0.50

**Table 8 ijerph-20-00728-t008:** Comparison of Factors’ Fit.

	χ^2^	DF	RMSEA	RMSEA 90% CI
2	2343.11	474	0.092	0.086–0.094
3	2132.22	564	0.086	0.079–0.090
4	19,884.34	662	0.078	0.067–0.081
5	1632.23	577	0.073	0.068–0.079
6	1488.34	559	0.071	0.068–0.075

**Table 9 ijerph-20-00728-t009:** Results of ESEM Analysis.

	Items			Sympathy	Coach Support	Care	Parents Support
26	Try to understand the thoughts and feelings of your colleagues.			0.754 *	0.033	0.330	0.287
16	I have a colleague who grants my request unconditionally.			0.920 *	0.092	0.112	0.199
25	When a colleague or friend gets sick, I feel pain too.			0.881 *	0.179	−0.083	−0.101
9	I have a leader who encourages me when I am having a hard time.			0.202	0.883 *	−0.021	0.132
12	There are leaders who want to do their best			0.194	0.911 *	0.111	0.077
14	I have a leader who will help me like my family			0.105	0.821 *	0.1330.	0.110
6	There are leaders who praise you when you do a task well.			−0.044	0.764 *	−0.093	0.099
27	Participate in setting the rules for the team			−0.100	0.055	0.665 *	0.221
28	Take on important tasks in the team.			−0.049	0.211	0.702 *	0.119
29	Participate in helping co-workers or those in need around them.			−0.010	0.204	0.800 *	−0.099
23	I often offer opinions for the development of myself and the team.			0.104	0.104	0.788 *	0.096
24	Never get discouraged when you lose a match or have a hard time			0.009	0.009	0.744 *	0.199
35	Never get discouraged when you lose a match or have a hard time			0.038	0.032	−0.108	0.792 *
27	Parents are very interested in athletes and school life			0.066	−0.037	0.021	0.662 *
26	My parents understand my feelings well.			0.047	−0.100	0.077	0.782 *
28	I work hard to make my dream come true.			0.091	−0.133	0.013	0.691 *
41	I work hard to make my dream come true.			0.044	−0.077	0.053	0.522 *
44	No matter what difficulties, I will surely achieve my dreams			0.058	−0.044	0.048	0.613 *
		sympathy	leader support	care		parent support/effort	
	sympathy	1					
	coach support	0.334	1				
	care	0.402	0.299	1			
	parent support/effort	0.332	0.402	0.377		1	

χ^2^ = 771.32, df = 338, CFI = 0.910, RMSEA = 0.062; * 0.05.

**Table 10 ijerph-20-00728-t010:** Results of ML CFA.

Factor		SC	S.E.	CR	AVE
sympathy	26	0.653	0.044	0.972	0.931
	16	0.867	0.045		
	25	0.778	0.042		
coach support	9	0.703	0.052	0.806	0.912
	12	0.756	0.053		
	14	0.774	0.054		
	6	0.737	0.052		
care	27	0.626	0.056	0.976	0.895
	28	0.499	0.052		
	29	0.804	0.065		
	23	0.825	0.066		
	24	0.798	0.064		
parent support/effort	35	0.705	0.059	0.961	0.887
	27	0.729	0.058		
	26	0.577	0.051		
	28	0.534	0.055		
	41	0.739	0.062		
	44	0.645	0.047		
		sympathy	leader support	care	parent support/effort
sympathy		1			
coach support		0.392	1		
care		0.322	0.344	1	
parent support/effort		0.401	0.498	0.332	1
**χ^2^**		df	RMSEA		TLI
802.2		407	0.0650		0.901

**Table 11 ijerph-20-00728-t011:** Response and item reliability.

	Factor	SEP	Rel.	Infit	Outfit
RR	sympathy	3.80	0.92	1.01	1.00
	coach support	3.65	0.90	1.03	1.01
	care	4.01	0.91	0.99	1.00
	parent support/effort	3.92	0.90	1.01	0.99
IR	sympathy	3.88	0.91	1.01	1.00
	coach support	3.72	0.91	1.00	0.99
	care	4.07	0.88	0.99	0.98
	parent support/effort	3.98	0.90	1.03	1.01

Note. RR: response reliability, IR: item reliability.

**Table 12 ijerph-20-00728-t012:** Results of verification of measurement equivalence by gender.

	X^2^	∆X^2^	df	*p*	∆df	RMSEA
Unconstrained model	924.33		443			0.041
Factor coefficient same constraint	956.44	32.11	465	0.111	16	0.041
Covariance equal constraint	980.95	56.62	484	0.133	41	0.043
Factor coefficient/Covariance/Error variance	988.96	64.63	492	0.108	49	0.043

## Data Availability

The data presented in this study are available on request from the first author. The data are not publicly available due to privacy issues.

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
