# Peer review of "Developing a Scale Using Item Response Theory of the Self-Resilience in Taekwondo Players"

_ijerph, 2022, doi:10.3390/ijerph20010728_

Round 1

Reviewer 1 Report

1. In the literature review section, it is suggested to increase the theoretical basis of the scale dimension construction

2. The age of the participants needs to be reported

3. Table 7 also contains non English, which needs to be modified

4. It is suggested to increase the test of retest reliability and criterion validity

Author Response

Before revised 1 In the literature review section, it is suggested to increase the theoretical basis of the scale dimension construction 2 The age of the participants needs to be reported 3 Table 7. non English After revised 1. The theoretical part of the scale was further explained. The new concept of ego-resiliency divides the concept into two dimensions—intrinsic and extrinsic—and ~ failing to provide an accurate empirical measurement of ego-resilience in individuals[10]. 2. Group A : 22.3 . Group B : 23.7 3. Modified

Reviewer 2 Report

Dear, authors,

The study is very simple and easy to understand, developing a new scale for taekwondo.

However, there are comments for the study.

The major points noted are the following.

(1) The participants are athletes from nine different countries. In addition, there are athletes whose native language is not English. Are the participants correctly understanding and responding to the English-only scale?

(2) When creating the scale, the participants were randomly divided, but it is necessary to verify the validity of the scale by dividing the participants into two groups: those whose native language is English and those whose native language is not English.

(3) The participants are athletes from 9 different countries, but what is the composition of the participants?

(4) There are several different ways of listing citations after L128. Please review all of them.

(5) The criteria used to validate are not consistent. The results using CFI are described in the figure, but not in the text. Please review them again.

(6) Please correct the use of a language other than English in table7.

Please review the manuscript from style to method and submit it again.

Best,

Author Response

A reply to the reviewer's opinion is attached and sent.
